# Neutrophil–Lymphocyte Ratio in Fibromyalgia and Axial Spondyloarthritis: A Potential Biomarker for Diagnosis and Disease Activity

**DOI:** 10.3390/biomedicines13061497

**Published:** 2025-06-18

**Authors:** Miriam Almirall, Esther Espartal, Xabier Michelena, Carlos Suso-Ribera, Mayte Serrat, Sara Marsal, Alba Erra

**Affiliations:** 1Department of Rheumatology, Hospital Universitari Vall d’Hebron, 08035 Barcelona, Spain; esther.espartal@vallhebron.cat (E.E.); xmichelena@gencat.cat (X.M.); carlos.suso@vallhebron.cat (C.S.-R.); mayte.serrat@vallhebron.cat (M.S.); sara.marsal@vallhebron.cat (S.M.); alba.erra@vallhebron.cat (A.E.); 2Rheumatology Research Group, Vall d’Hebron Research Institute, 08035 Barcelona, Spain; 3Digitalization for the Sustainability of the Healthcare System (DS3), Catalan Health Service, 08007 Barcelona, Spain

**Keywords:** Neutrophil–Lymphocyte Ratio, Fibromyalgia, Axial Spondyloarthritis, biomarker, inflammation

## Abstract

**Objective**: The Neutrophil–Lymphocyte Ratio (NLR) has been proposed as an inflammatory biomarker in several diseases, including Fibromyalgia, with controversial results. The objectives of this study were to: (1) compare NLR values among participants with Fibromyalgia, Axial Spondyloarthritis, and healthy controls; (2) assess the relationship between NLR and disease activity; and (3) establish diagnostic and activity cut-off values. **Methods**: A total of 112 age and gender-matched participants were included in each group. NLR values were compared between groups, correlations with disease activity were analyzed, and cut-off values were calculated using Receiver Operating Characteristic (ROC) curves. **Results**: The NLR was significantly higher in Fibromyalgia patients compared with healthy controls (1.8 ± 0.5 vs. 1.4 ± 0.2; *p* < 0.001) and in Axial Spondyloarthritis patients compared with both Fibromyalgia patients (2.1 ± 0.3 vs. 1.8 ± 0.5; *p* < 0.001) and healthy controls (2.1 ± 0.3 vs. 1.4 ± 0.2; *p* < 0.001). Within disease groups, the NLR was also significantly higher in patients with severe Fibromyalgia (FIQ ≥ 59) compared with non-severe cases (1.9 ± 0.5 vs. 1.7 ± 0.4; *p* = 0.008) and in patients with high/very high Axial Spondyloarthritis activity compared with those with low/inactive disease (2.3 ± 0.3 vs. 1.9 ± 0.2; *p* < 0.001). ROC analysis identified the NLR cut-off values of 1.54 for Fibromyalgia diagnosis, 1.64 for severe disease, 1.61 for Axial Spondyloarthritis diagnosis and 1.95 for high/very high disease activity. **Conclusions**: The NLR may serve as a cost-effective, rapid, and accessible biomarker for establishing diagnosis and disease activity in Axial Spondyloarthritis and, to a lesser extent, in Fibromyalgia. Further research is needed to validate these findings and explore NLR’s role alongside other inflammatory markers.

## 1. Introduction

The Neutrophil–Lymphocyte Ratio (NLR) is a widely accessible and cost-effective inflammatory biomarker derived from a simple blood count. Its significance arises from observations linking a higher absolute neutrophil count to an increased cardiovascular risk and a lower absolute lymphocyte count to a reduced survival [1,2].

It has been associated with cardiovascular risk [3] and immune-mediated inflammatory diseases [4,5,6].

Of particular interest is a study that analyzed data from five randomized clinical trials involving over 60,000 participants. Baseline NLR predicted cardiovascular risk and all-cause mortality and was reduced by an inflammation-targeting treatment, interleukin-1β inhibitor canakinumab [3].

Beyond cardiovascular disease, the NLR has also been proposed as a biomarker of systemic inflammation in various immune-mediated inflammatory diseases. In Rheumatoid Arthritis, Systemic Lupus Erythematosus, and Axial Spondyloarthritis, NLR has been associated with diagnosis, disease activity, and even treatment response [4,5,6]. Given the inflammatory basis of these conditions, the role of the NLR as a disease activity marker is well established.

However, its relevance in Fibromyalgia remains unclear [7,8,9,10,11]. Unlike classic immune-mediated inflammatory diseases, Fibromyalgia is not primarily driven by peripheral inflammation but rather by central sensitization, neuroinflammatory phenomena, and dysregulation of the autonomic and immune systems [12,13,14,15,16,17]. Despite this, some studies have reported elevated NLR values in patients with Fibromyalgia compared with healthy controls [7,8], suggesting that low-grade inflammation or stress-related immune dysregulation may contribute to disease pathology. Conversely, others have found no significant differences in the NLR between Fibromyalgia patients and healthy controls [9]. Furthermore, only one study to date has examined whether the NLR changes in response to treatment, showing a significant decrease in the NLR after three months of duloxetine therapy [10]. Given these conflicting findings, further research is needed to clarify the role of the NLR in Fibromyalgia and its potential utility as a biomarker of disease severity.

The primary objective of the present study was to evaluate differences in NLR values between patients with Fibromyalgia, Axial Spondyloarthritis, and healthy controls. The secondary objectives were to assess the relationship between NLR and disease activity and to establish diagnostic and severity/activity cut-off values.

Based on previous studies suggesting a potential link between the NLR and inflammatory or stress-related conditions, we hypothesize the following:(1)Patients with Fibromyalgia and Axial Spondyloarthritis have significantly higher NLR values than healthy controls, reflecting a heightened inflammatory or stress-related response.(2)Axial Spondyloarthritis patients exhibit the highest NLR values due to the well-established inflammatory nature of the disease, whereas Fibromyalgia patients will show intermediate NLR values between healthy controls and Axial Spondyloarthritis patients.(3)Higher NLR values are associated with greater disease severity in both Fibromyalgia and Axial Spondyloarthritis, as measured by disease-specific severity/activity scores.(4)Specific NLR cut-off values can be established to differentiate patients from controls and to classify disease severity/activity in both conditions.

## 2. Materials and Methods

### 2.1. Study Design

This is a single-center, cross-sectional, observational study conducted at a university hospital between September and December 2024.

### 2.2. Participants

Participants were recruited from the Rheumatology Department at the Vall d’Hebron University Hospital and categorized into three groups:(1)Fibromyalgia patients: Diagnosed based on clinical criteria and recruited from a specialized Fibromyalgia unit. These patients did not meet the criteria for any inflammatory joint disease.(2)Axial Spondyloarthritis patients: Diagnosed based on clinical and imaging criteria and recruited from a specialized unit. These patients did not meet the criteria for Fibromyalgia.(3)Healthy controls: Recruited from a complex low back pain unit. These individuals had no clinical diagnosis of Fibromyalgia or inflammatory joint disease and had undergone a recent blood test (within six months of study enrollment).

There were no other exclusion criteria due to the cross-sectional design of the study, which did not evaluate the efficacy of any intervention.

All participants had a complete blood count, performed at the tertiary hospital and measured using the same equipment.

Both healthy controls and patients with Axial Spondyloarthritis were age and gender-matched with the selected patients with Fibromyalgia to ensure comparability.

### 2.3. Measures

Demographic, clinical, and laboratory parameters were collected for all participants.
-Common variables (all groups):
Age, gender, and Body Mass Index (BMI);Cardiovascular risk factors: Arterial hypertension, dyslipidemia, diabetes mellitus, history of acute myocardial infarction, and history of stroke;Medication use: Regular Non-Steroidal Anti-Inflammatory Drug (NSAID) consumption (>3 times per week);C-Reactive Protein (CRP) levels (mg/dL). Measured at the tertiary hospital, using the same equipment and kit manufacturer.-Fibromyalgia and Axial Spondyloarthritis groups only:
Disease duration: Time since symptom onset and time since diagnosis;Disease activity/severity scores;Fibromyalgia: Revised Fibromyalgia Impact Questionnaire (FIQR) [18].Axial Spondyloarthritis: Axial Spondyloarthritis Disease Activity Score (ASDAS) [19] and Bath Ankylosing Spondylitis Disease Activity Index (BASDAI) [20].-Axial Spondyloarthritis group only:
Current treatment with Disease-Modifying Antirheumatic Drugs (DMARDs): Conventional synthetic (csDMARDs), biological (bDMARDs), and targeted synthetic (tsDMARDs).

### 2.4. Statistical Analysis

-Group comparisons: Differences between groups were calculated using the Kruskal–Wallis test for quantitative variables and the Chi-square test or Fisher’s exact test for categorical variables (depending on expected frequencies).-Correlation analysis: The correlation between the NLR and disease severity/activity scores (FIQR, ASDAS, and BASDAI) was calculated using Pearson’s correlation coefficients.-Cut-off determination: The NLR cut-off values for diagnosis and disease activity were calculated using Receiver Operating Characteristic (ROC) curves.

For all analyses, *p* values < 0.05 were considered statistically significant.

All analyses were performed with the statistical program “R” (version 4.3.3, 2024-02-29 ucrt, Copyright © 2015 The R Foundation for Statistical Computing).

### 2.5. Ethical Considerations

The study was conducted in accordance with the Declaration of Helsinki and approved by the Vall d’Hebron University Hospital Ethical Committee (registration code PR(AG)337/2024).

Patient consent was waived in accordance with current legislation (LOPDGDD, 3/2018), due to the observational design of the study, the impracticality of carrying it out without waiver, the security protection provided through data pseudonymization and the absence of additional risk to participants.

## 3. Results

A total of 112 patients with Fibromyalgia, 112 patients with Axial Spondyloarthritis, and 112 healthy controls were included in the study. All the participants were age and gender-matched.

The main demographic, clinical, and analytical variables of the three groups of participants and their comparisons are shown in Table 1.

There were statistically significant differences in mean NLR values between patients with Fibromyalgia and healthy controls (1.8 ± 0.5 vs. 1.4 ± 0.2; *p* < 0.001), between Fibromyalgia and Axial Spondyloarthritis patients (1.8 ± 0.5 vs. 2.1 ± 0.3; *p* < 0.001), and between Axial Spondyloarthritis patients and healthy controls (2.1 ± 0.3 vs. 1.4 ± 0.2; *p* < 0.001).

Similarly, significant differences were detected in NSAID consumption between Fibromyalgia and Axial Spondyloarthritis patients (29, 25.9%, vs. 58, 51.8%; *p* < 0.001) and between Axial Spondyloarthritis patients and healthy controls (58, 51.8% vs. 28, 25%; *p* < 0.001). A statistically significant difference was also observed in the mean CRP value between Fibromyalgia and Axial Spondyloarthritis patients (0.5 ± 0.2 vs. 0.6 ± 0.3; *p* = 0.01). There were no other statistically significant differences between the rest of the variables among the groups.

Among Fibromyalgia patients, 83 (74.1%) had severe disease (FIQR score ≥ 59), while 57 (50.9%) of Axial Spondyloarthritis patients had high/very high disease activity (ASDAS ≥ 2.1).

In Fibromyalgia patients, the NLR was positively correlated with CRP and FIQR scores, though with a small effect size (r = 0.14 and r = 0.15, respectively; Table 2). In Axial Spondyloarthritis patients, the NLR demonstrated a stronger correlation with CRP (r = 0.38) and was significantly correlated with ASDAS and BASDAI (r = 0.62 and r = 0.57, respectively), with medium to large effect sizes (Table 2).

Stratification by disease severity/activity revealed significant differences in mean NLR values: Fibromyalgia patients with severe disease (FIQR ≥ 59) had higher NLR values than those with non-severe disease (1.9 ± 0.5 vs. 1.7 ± 0.4; *p* = 0.008). Similarly, Axial Spondyloarthritis patients with high/very high disease activity (ASDAS ≥ 2.1) had significantly higher NLR values than those with low/inactive disease activity (2.3 ± 0.3 vs. 1.9 ± 0.2; *p* < 0.001; Table 3 and Table 4).

Receiver Operating Characteristic (ROC) analysis determined the optimal NLR cut-off values for diagnosis and disease severity/activity prediction (Table 5).

For Fibromyalgia diagnosis, the NLR threshold was 1.54 (AUC = 0.81, 95% CI = 0.76–0.87), with a sensitivity of 70.54% and specificity of 82.14%. (Figure 1A). For Axial Spondyloarthritis diagnosis, the optimal cut-off was 1.61 (AUC = 0.97, 95% CI = 0.95–0.99), with a sensitivity of 98.21% and specificity of 91.96% (Figure 1B). Additionally, the NLR cut-off value for predicting severe Fibromyalgia (FIQR ≥ 59) was 1.64 (AUC = 0.67, 95% CI = 0.55–0.78), with a sensitivity of 66.27% and specificity of 62.07% (Figure 1C). For high/very high disease activity in Axial Spondyloarthritis (ASDAS ≥ 2.1), the cut-off was 1.95 (AUC = 0.89, 95% CI = 0.84–0.95), with a sensitivity of 92.98% and specificity of 77.78% (Figure 1D).

## 4. Discussion

To the best of our knowledge, this is the first study to assess differences in NLR values among Fibromyalgia patients, healthy controls, and Axial Spondyloarthritis patients, as well as to explore their relationship with activity and establish cut-off values for both diagnosis and disease activity. Our findings revealed significant differences in mean NLR values across all groups. Moreover, the NLR was correlated with CRP and disease activity scores in both Fibromyalgia and Axial Spondyloarthritis. The established cut-off values demonstrated high sensitivity and specificity for both diagnostic and disease activity predictions.

In our study, we observe statistically significant differences in the mean NLR values between the three groups of participants and between those with severe/active and non-severe/non-active Fibromyalgia and Axial Spondyloarthritis. We also found that NLR values correlated with CRP and disease activity scores in two diseases. This is the first study to evaluate differences in NLR values between Fibromyalgia and Axial Spondyloarthritis, although there are previous reports comparing the two diseases and healthy controls that also evaluate the relationship with disease activity scores and the response to treatments with different results [6,7,8,9,10,11,21,22,23,24,25,26].

The role of the NLR in Fibromyalgia remains controversial, with previous studies yielding conflicting results. Some studies have reported higher NLR values in Fibromyalgia patients compared with healthy controls, suggesting a potential role for low-grade inflammation [7,8]. Others, however, have found no significant differences [9,10].

Additionally, studies investigating the correlation between the NLR and Fibromyalgia severity, as measured by the FIQR, have generally failed to establish a meaningful association [8,11]. The discrepancies between studies could be attributed to differences in sample characteristics, such as BMI, comorbidities, disease severity, and drug use. It has been argued that BMI and metabolic syndrome in particular can influence different inflammatory biomarkers, such as high-sensitivity CRP or the NLR [27,28,29]. Although BMI was not significantly different between groups in our study, its potential role in modulating the NLR levels warrants further investigation.

Drug use is another factor that may contribute to the discrepancies in findings. NSAIDs and other medications known to impact inflammatory markers have been used at varying rates across studies potentially influencing NLR values [30]. In our study, NSAID consumption was significantly higher in the Axial Spondyloarthritis group compared with Fibromyalgia patients and healthy controls. This may partially explain the more pronounced NLR values observed in Axial Spondyloarthritis, aside from the disease’s inherent inflammatory activity.

In a study by Gökmen et al., involving patients with Ankylosing Spondylitis, those receiving anti-TNF therapy had lower NLR values than those treated with NSAIDs [31]. These results were likely influenced by both the greater reduction in inflammatory activity achieved with biologic therapy and the possible effects of NSAIDs on NLR.

Alternatively, elevations in the NLR among Fibromyalgia patients may only become apparent in those with more severe manifestations of the disease. Significant differences in NLR values were observed between severe (FIQR ≥ 59) and non-severe Fibromyalgia patients (*p* = 0.008), suggesting a potential threshold effect. Studies with a higher proportion of mild-to-moderate Fibromyalgia cases may have failed to detect an association between the NLR and Fibromyalgia severity. Future research should investigate whether specific clinical subtypes of Fibromyalgia—such as those with higher inflammatory burden or central sensitization features—are more likely to exhibit an elevated NLR.

Despite the obvious limitations of activity scores, primarily the subjectivity of assessments and the influence of external factors, a standardized stratification of Fibromyalgia patients based on disease severity is essential. This allows us to investigate whether different patient subtypes, such as those with a higher inflammatory burden or more evidence of central sensitization, are more likely to have higher values of both NLR and other inflammatory biomarkers.

We used the FIQ severity categories established in the study by Bennet et al., based on three placebo-controlled clinical treatment trials [32].

In addition to baseline differences, the NLR changes in response to treatment have also been evaluated in Fibromyalgia. In particular, one report demonstrated a significant reduction in the NLR following duloxetine therapy, suggesting that the NLR may be influenced by treatment interventions [10]. While promising, this finding has yet to be widely replicated; therefore, the role of the NLR as a dynamic marker of treatment response in Fibromyalgia remains unclear and should be further explored in longitudinal studies.

There are data on the involvement of the immune system in the etiopathogenesis of Fibromyalgia, such as the recent findings of the presence of antibodies against satellite glial cells of the dorsal root ganglia compared with healthy controls and its relationship with the severity of the disease [13,14,15]. This involvement of the immune system, together with other previous data on the existence of neuroinflammatory phenomena [16,17], could explain the increase in the NLR, a biomarker of systemic inflammation [4,5,6], compared with healthy controls and its relationship with disease activity observed in our study and in previous reports [7,8,9]. While acknowledging this, further research is needed to clarify the role of inflammation in Fibromyalgia and to determine whether the NLR can reliably reflect disease activity.

In Axial Spondyloarthritis, where inflammation is well established [33], multiple studies have confirmed significantly higher NLR values compared with healthy controls and a positive association with disease activity [6,21,22,23,24,25,26]. The utility of the NLR as a biomarker for disease persistence and treatment response has also been demonstrated, with levels shown to decrease following biological therapy [21,24,25,26]. The relationship between the NLR and disease activity in Axial Spondyloarthritis has also been explored in several studies, with some inconsistencies. A recent meta-analysis of 11 studies involving 820 patients with Ankylosing Spondylitis and 743 age and gender-matched controls found that while six studies reported significantly higher NLRs in Ankylosing Spondylitis, the pooled mean difference was only 0.38 and not statistically significant [6]. The authors suggested that variations in disease activity among participants might explain these discrepancies [6]. Consistent with this idea, some studies have found stronger associations when comparing active versus inactive disease. For example, Kucuk et al. [22] reported significantly higher NLR values in active Ankylosing Spondylitis patients (BASDAI ≥ 4) compared with those with mild disease (*p* = 0.001). Similarly, Zeb et al. observed that the NLR was significantly higher in active disease compared with inactive cases (*p* < 0.01) [23]. These findings align with our study, in which we observed significantly higher NLR values in patients with high/very high disease activity (ASDAS ≥ 2.1) compared with those with low/inactive disease (*p* < 0.001). This reinforces the potential role of the NLR as a marker of inflammatory burden in Axial Spondyloarthritis, particularly for assessing disease activity.

To stratify patients based on disease activity, we used ASDAS because it has demonstrated improved measurement properties compared with BASDAI in Axial Spondyloarthritis [34].

Regarding the response to biological treatments, Coşkun et al. [21] found that the NLR decreased significantly after three months of anti-TNF therapy (2.67 ± 1.17 vs. 1.8 ± 0.7; *p* < 0.001), correlating with BASDAI, Erythrocyte Sedimentation Rate (ESR), and CRP. Similarly, a 2022 post hoc analysis of 19 phase III/IV secukinumab trials confirmed that the NLR rapidly decreased after 12–16 weeks of treatment and remained low after one year [26]. Our findings align with the previous reports, further supporting the NLR as a potential biomarker of inflammatory activity in Axial Spondyloarthritis. However, due to the cross-sectional nature of our study, we could not evaluate the NLR’s role in treatment response.

In addition to disease activity and drug use, comorbidities represent other potential confounding factors that could cause elevated NLR levels in patients with Axial Spondyloarthritis. As discussed in the context of Fibromyalgia, BMI and metabolic syndrome can increase the levels of inflammatory biomarkers, including the NLR [27,28,29]. In our study, BMI and comorbidities associated with metabolic syndrome—such as dyslipidemia and diabetes mellitus—did not differ significantly among the three participant groups. This suggests that these factors likely did not contribute meaningfully to the observed differences in NLR values between groups. We also established cut-off values for diagnosis and disease severity/activity: 1.54 for Fibromyalgia diagnosis, 1.61 for Axial Spondyloarthritis diagnosis, 1.64 for severe Fibromyalgia (FIQR ≥ 59), and 1.95 for high/very high disease activity in Axial Spondyloarthritis (ASDAS ≥ 2.1). Importantly, while our findings indicate that the NLR is significantly elevated in Fibromyalgia compared with healthy controls, its diagnostic accuracy (AUC = 0.81, sensitivity = 70.54%, specificity = 82.14%) was lower than that observed for Axial Spondyloarthritis (AUC = 0.97, sensitivity = 98.21%, specificity = 91.96%). This suggests that while the NLR may be a useful biomarker in Fibromyalgia, it has a greater diagnostic value in Axial Spondyloarthritis. The weaker association in Fibromyalgia may be due to its complex and heterogeneous pathophysiology, involving central sensitization, autoimmunity, autonomic dysfunction, and low-grade inflammation [12,13,14,15,16,17].

Prior studies have reported a Fibromyalgia diagnostic cut-off value of 1.6, with lower sensitivity and specificity than those found in our study, reinforcing our findings that while the NLR may have some diagnostic value, its performance is limited in this condition [7]. In contrast, our Axial Spondyloarthritis cut-off value of 1.95 for high/very high disease activity (ASDAS ≥ 2.1) was similar to a previously reported value (1.91) but demonstrated superior sensitivity (92.98% vs. 69%) and specificity (77.78% vs. 54%) [22]. These results highlight the potential clinical utility of the NLR in Axial Spondyloarthritis, while also underscoring the need for further research into its role in Fibromyalgia. A more promising role for the NLR in Fibromyalgia may be in monitoring disease severity rather than diagnosis. In our study, the NLR was significantly higher in patients with severe Fibromyalgia (FIQR ≥ 59) compared with those with milder forms (*p* = 0.008). However, the correlation between the NLR and FIQR scores was weak (r = 0.15), suggesting that while the elevated NLR may reflect heightened disease burden, it should not be used in isolation to assess severity. One possible explanation is that Fibromyalgia is a heterogeneous condition, with inflammation playing a role only in a subset of patients [12,13,14,15,16,17]. Alternatively, the NLR may be influenced by factors unrelated to Fibromyalgia severity, such as stress, autonomic dysfunction, metabolic status, BMI, comorbidities, disease severity, or drug use [35], reducing its predictive reliability. Future studies should investigate the influence of external factors and the value of combining the NLR with other inflammatory or immune-related biomarkers (e.g., platelet-to-lymphocyte ratio, cytokine levels, or CRP-adjusted NLR) to develop a more comprehensive index of disease activity. Additionally, prospective studies should examine whether changes in the NLR over time correlate with symptom fluctuations and treatment response, as suggested by recent evidence showing a decrease in the NLR after duloxetine therapy [10], which may provide better insight into its clinical utility.

Further longitudinal studies are needed in both diseases, Axial Spondyloarthritis and Fibromyalgia, to evaluate the NLR’s role as a dynamic biomarker of response to different therapies.

The NRL was positively correlated with CRP in both diseases, although with a smaller effect size in Fibromyalgia patients (r = 0.14) compared with Axial Spondyloarthritis patients (r = 0.38). These findings may be attributed to the fact that Axial Spondyloarthritis is a widely known and extensively studied disease in which inflammation plays a central role [33]. However, as previously mentioned, further research is needed to clarify the role of inflammation in Fibromyalgia. Although Fibromyalgia is a highly heterogeneous condition that has not traditionally been associated with elevated levels of acute phase reactants, a study by Singh et al. found that increased disease severity was linked to higher levels of CRP and ESR [36]. The main limitations of our study include its observational, cross-sectional, and single-center design, the relatively small sample size and the selection of healthy controls from a complex low back pain unit, even though none met the criteria for Fibromyalgia or any inflammatory rheumatic diseases. Despite these limitations, our study provides valuable insights into the potential role of NLR as a biomarker in Fibromyalgia and especially in Axial Spondyloarthritis.

In future studies aimed at validating these findings and exploring the NLR’s role in these diseases, it will be preferable to select a healthy control group without chronic pain and other rheumatic conditions. Although low back pain patients were evaluated by rheumatologists who ruled out Fibromyalgia and inflammatory rheumatic diseases, the presence of subclinical inflammation could not be completely excluded, which may introduce selection bias. A control group without chronic pain would represent the general healthy population more accurately and allow for a more appropriate comparison with the patient groups.

## 5. Conclusions

In our study, the NLR was significantly higher in patients with Axial Spondyloarthritis and Fibromyalgia than in healthy controls, as well as in those with severe/active disease compared with non-severe/non-active cases. We also observed that NLR values correlated with CRP levels and disease activity scores in both conditions. Moreover, specific NRL cut-off values demonstrated high sensitivity and specificity for both diagnostic and disease activity predictions. All results were stronger and more consistent in Axial Spondyloarthritis than in Fibromyalgia.

These findings suggest that the NLR may serve as a promising, cost-effective, rapid, and accessible biomarker for establishing diagnosis and disease activity in Axial Spondyloarthritis and, to a lesser extent, in Fibromyalgia. Further large-scale, multicenter, and prospective studies are needed to validate these findings and to explore the NLR’s potential in monitoring treatment response, either alone or in combination with other inflammatory biomarkers.

## Figures and Tables

**Figure 1 biomedicines-13-01497-f001:**
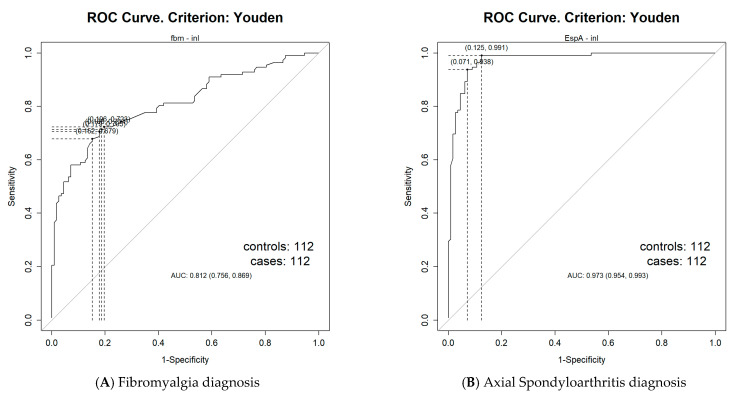
Receiver Operating Characteristic curves for determining the Neutrophil–Lymphocyte Ratio cut-off values for predicting diagnosis and disease activity. (**A**) For Fibromyalgia diagnosis, the Neutrophil–Lymphocyte Ratio (NLR) cut-off value, obtained from Youden’s index, was 1.54 (Area Under the curve (AUC) = 0.81, 95% Confidence Interval (CI) = 0.76–0.87), with a sensitivity of 70.54% and specificity of 82.14%. (**B**) For Axial Spondyloarthritis diagnosis, the Neutrophil–Lymphocyte Ratio (NLR) cut-off value, obtained from Youden’s index, was 1.61 (Area Under the curve (AUC) = 0.97, 95% Confidence Interval (CI) = 0.95–0.99), with a sensitivity of 98.21% and specificity of 91.96%. (**C**) For severe Fibromyalgia, the Neutrophil–Lymphocyte Ratio (NLR) cut-off value, obtained from Youden’s index, was 1.64 (Area Under the curve (AUC) = 0.67, 95% Confidence Interval (CI) = 0.55–0.78), with a sensitivity of 66.27% and specificity of 62.07%. (**D**) For high/very high disease activity in Axial Spondyloarthritis, the Neutrophil–Lymphocyte Ratio (NLR) cut-off value, obtained from Youden’s index, was 1.95 (Area Under the curve (AUC) = 0.89, 95% Confidence Interval (CI) = 0.84–0.95), with a sensitivity of 92.98% and specificity of 77.78%.

**Table 1 biomedicines-13-01497-t001:** Demographic, clinical, and analytical variables in the three groups of participants. Statistically significant *p* values are marked in bold.

Variables	FM (n = 112)	HC (n = 112)	axSpA (n = 112)	*p* Value FM-HC	*p* Value FM-AxSpA	*p* Value AxSpA-HC
Age, mean (SD)	49.4 (±7.5)	49.1 (±7.7)	49.2 (±7.4)	0.89	0.83	0.97
Women, n (%)	94 (83.9%)	94 (83.9%)	94 (83.9%)	0.99	0.99	0.99
NLR, mean (SD)	1.8 (±0.5)	1.4 (±0.2)	2.1 (±0.3)	**<0.001**	**<0.001**	**<0.001**
BMI, mean (SD)	27.3 (±5.6)	26.1 (±4.1)	26.2 (±4.6)	0.21	0.22	0.96
AHT, n (%)	20 (17.9%)	21 (18.8%)	20 (17.9%)	0.99	0.98	0.98
DLP, n (%)	25 (22.3%)	18 (16.1%)	31 (27.7%)	0.31	0.44	0.05
DM, n (%)	11 (9.8%)	7 (6.2%)	10 (8.9%)	0.46	0.98	0.61
AMI, n (%)	1 (0.9%)	1 (0.9%)	2 (1.8%)	0.99	0.98	0.98
Stroke, n (%)	1 (0.9%)	0 (0.0%)	2 (1.8%)	0.98	0.98	0.50
NSAIDs, n (%)	29 (25.9%)	28 (25%)	58 (51.8%)	0.98	**<0.001**	**<0.001**
CRP (mg/dL), mean (SD)	0.5 (±0.2)		0.6 (±0.3)		**0.01**	
Duration sso, (years), mean (SD)	11.7 (±7.2)		17.7 (±8.5)			
Duration sdg, (years), mean (SD)	7.2 (±6.1)		14.3 (±8.3)			
FIQR, mean (SD)	71.8 (±14.3)					
FIQR ≥ 59, n (%)	83 (74.1%)					
ASDAS, mean (SD)			2.3 (±1.0)			
ASDAS ≥ 2.1, n (%)			57 (50.9%)			
BASDAI, mean (SD)			35.7 (±21.3)			
csDMARD, n (%)			22 (19.6%)			
bDMARD, n (%)			70 (62.5%)			
tsDMARD, n (%)			14 (12.5%)			

FM = Fibromyalgia, HC = Healthy Controls, AxSpA = Axial Spondyloarthritis, n = Number, NLR = Neutrophil–Lymphocyte Ratio, BMI = Body Mass Index, AHT = Arterial Hypertension, DLP = Dyslipidemia, DM = Diabetes Mellitus, AMI = Acute Myocardial Infarction, NSAIDs = Non-Steroidal Anti-Inflammatory Drugs, CRP = C-Reactive Protein, mg = milligrams, dL = deciliter, sso = since symptom onset, sdg = since diagnosis, FIQR = Revised Fibromyalgia Impact Questionnaire, ASDAS = Axial Spondyloarthritis Disease Activity Score, BASDAI = Bath Ankylosing Spondylitis Disease Activity Index, csDMARD = conventional synthetic Disease-Modifying Antirheumatic Drugs, bDMARD = biological Disease-Modifying Antirheumatic Drugs, tsDMARDs = targeted synthetic Disease-Modifying Antirheumatic Drugs.

**Table 2 biomedicines-13-01497-t002:** Correlation of activity-related variables and the Neutrophil–Lymphocyte Ratio in patients with Fibromyalgia and Axial Spondyloarthritis.

Variables	CC with NLR in FM	CC with NLR in AxSpA
CRP	0.14	0.38
FIQR	0.15	
ASDAS		0.62
BASDAI		0.57

CC = Correlation Coefficient, NLR = Neutrophil–Lymphocyte Ratio, FM = Fibromyalgia, AxSpA = Axial Spondyloarthritis, CRP = C-Reactive Protein, FIQR = Revised Fibromyalgia Impact Questionnaire, ASDAS = Axial Spondyloarthritis Disease Activity Score, BASDAI = Bath Ankylosing Spondylitis Disease Activity Index.

**Table 3 biomedicines-13-01497-t003:** Differences in the Neutrophil–Lymphocyte Ratio values in patients with Fibromyalgia based on severity. Statistically significant *p* values are marked in bold.

	FIQR ≥ 59 (Severe FM)	FIQR < 59 (Moderate or Mild FM)	*p* Value
NLR value	1.9 (±0.5)	1.7 (±0.4)	**0.008**

NLR = Neutrophil–Lymphocyte Ratio, FIQR = Revised Fibromyalgia Impact Questionnaire, FM = Fibromyalgia.

**Table 4 biomedicines-13-01497-t004:** Differences in Neutrophil–Lymphocyte Ratio values in patients with Axial Spondyloarthritis based on activity. Statistically significant *p* values are marked in bold.

	ASDAS ≥ 2.1 (High/Very High Disease Activity)	ASDAS < 2.1 (Inactive/Low Disease Activity)	*p* Value
NLR value	2.3 (±0.3)	1.9 (±0.2)	**<0.001**

NLR = Neutrophil–Lymphocyte Ratio, ASDAS = Axial Spondyloarthritis Disease Activity Score.

**Table 5 biomedicines-13-01497-t005:** Neutrophil–Lymphocyte Ratio cut-off values associated with diagnosis and activity in Fibromyalgia and Axial Spondyloarthritis.

	FM Diagnosis	AxSpA Diagnosis	Severe FM	High/Very High Disease Activity in AxSpA
NLR cut-off value	1.54	1.61	1.64	1.95
	AUC = 0.81 (95% CI: 0.76–0.87)	AUC = 0.97 (95% CI: 0.95–0.99)	AUC = 0.66 (95% CI: 0.55–0.78)	AUC = 0.89 (95% CI: 0.84–0.95)
	SE = 70.54%	SE = 98.21%	SE = 66.27%	SE = 92.98%
	SP = 82.14%	SP = 91.96%	SP = 62.07%	SP = 77.78%

NLR = Neutrophil–Lymphocyte Ratio, FM = Fibromyalgia, AxSpA = Axial Spondyloarthritis, AUC = Area Under the Curve, CI = confidence Interval, SE = Sensitivity, SP = Specificity.

## Data Availability

The data that support the findings of this study are available from the corresponding author, upon reasonable request.

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
