# Peer review of "Neutrophil–Lymphocyte Ratio in Fibromyalgia and Axial Spondyloarthritis: A Potential Biomarker for Diagnosis and Disease Activity"

_biomedicines, 2025, doi:10.3390/biomedicines13061497_

Round 1
Reviewer 1 Report
Comments and Suggestions for Authors
Dear authors,
This manuscript addresses a significant gap in the current literature and could meaningfully contribute to clinical practice if its findings are replicated. The study design is solid, and the topic is relevant. However, several weaknesses should be addressed to improve the quality and credibility of the manuscript.
- Interpretation of the Results
- The discussion would benefit from a more critical reflection on the modest correlation coefficients between NLR and FIQR in fibromyalgia. Although statistically significant, the effect size is minimal and should be described accordingly.
- I recommend adding a paragraph that acknowledges the limited strength of this correlation and discusses alternative explanations for elevated NLR in fibromyalgia—such as metabolic syndrome, unrelated inflammation, or stress-related immune dysregulation.
- Potential Confounding Factors
- There is insufficient discussion on how comorbidities or medication use (e.g., NSAIDs, antidepressants) might have influenced NLR values.
- You should explicitly acknowledge this limitation and its implications. If feasible, consider including a sensitivity or subgroup analysis to assess potential confounding effects.
- Control Group Selection
- The use of participants from a “low back pain unit” as healthy controls could introduce bias, particularly if any had subclinical inflammation.
- Please include a brief discussion on how this population might not represent the general healthy population and suggest how future studies could better address this issue through broader sampling.
- Exclusion Criteria
- Please clarify the exclusion criteria used in participant selection to ensure sample purity and minimize potential sources of bias.
- Formatting and Presentation
- Expand all abbreviations upon first use in the main text for clarity.
- Figures and tables: Figure 1 (panels A–D) is referenced in the text but not shown in the document excerpt. Ensure that all figures are included, high-quality, well-labeled, and correctly cited in the final version.
- Punctuation and spacing around units (e.g., "mg/dL") must be standardized. Inconsistent formatting detracts from readability and gives the impression of a lack of attention to detail.
- Correct typographical errors, such as “17,.9%” in Table 1. A thorough proofreading of the entire manuscript is strongly recommended.
Author Response
Dear reviewer,
Thank you very much for your considerations and insightful comments that will help improve our manuscript.
We proceed to answer the reviewers' comments.
Reviewer’s comment
- Interpretation of the Results
The discussion would benefit from a more critical reflection on the modest correlation coefficients between NLR and FIQR in fibromyalgia. Although statistically significant, the effect size is minimal and should be described accordingly.
I recommend adding a paragraph that acknowledges the limited strength of this correlation and discusses alternative explanations for elevated NLR in fibromyalgia—such as metabolic syndrome, unrelated inflammation, or stress-related immune dysregulation.
Author’s response
We have expanded the paragraph in the Discussion section that addresses the weak association between NLR and FIQR, along with the potential influencing factors.
“In our study, NLR was significantly higher in patients with severe Fibromyalgia (FIQR≥59) compared to those with milder forms (p=.008). However, the correlation between NLR and FIQR scores was weak (r=.15), suggesting that while elevated NLR may reflect heightened disease burden, it should not be used in isolation to assess severity. One possible explanation is that Fibromyalgia is a heterogeneous condition, with inflammation playing a role only in a subset of patients [12-17]. Alternatively, NLR may be influenced by factors unrelated to Fibromyalgia severity, such as stress, autonomic dysfunction, metabolic status, BMI, comorbidities, disease severity or drugs use, reducing its predictive reliability. Future studies should investigate the influence of external factors and the value of combining NLR with other inflammatory or immune-related biomarkers (e.g., platelet-to-lymphocyte ratio, cytokine levels, or CRP-adjusted NLR) to develop a more comprehensive index of disease activity. Additionally, prospective studies should examine whether changes in NLR over time correlate with symptom fluctuations and treatment response, as suggested by recent evidence showing a decrease in NLR after duloxetine therapy [10], which may provide better insight into its clinical utility.”
Reviewer’s comment
- Potential Confounding Factors
There is insufficient discussion on how comorbidities or medication use (e.g., NSAIDs, antidepressants) might have influenced NLR values.
You should explicitly acknowledge this limitation and its implications. If feasible, consider including a sensitivity or subgroup analysis to assess potential confounding effects.
Author’s response
Following your recommendation, we have expanded the discussion on how comorbidities may influence NLR values in both Fibromyalgia and Axial Spondyloarthritis. Additionally, we have included two new references (22 and 23) that address the relationship between BMI, metabolic syndrome, and various inflammatory biomarkers, including NLR.
Reviewer’s comment
- Control Group Selection
The use of participants from a “low back pain unit” as healthy controls could introduce bias, particularly if any had subclinical inflammation.
Please include a brief discussion on how this population might not represent the general healthy population and suggest how future studies could better address this issue through broader sampling.
Author’s response
A paragraph has been added at the end of the Discussion section addressing the use of participants with lower back pain as healthy controls.
“In future studies aimed to validate these findings and to explore the NLR’s role in these diseases, it would be preferable to select healthy control group without chronic pain and other rheumatic conditions. Although the low back pain patients were evaluated by rheumatologists who ruled out Fibromyalgia and inflammatory rheumatic diseases, the presence of subclinical inflammation could not be completely excluded, which may introduce selection bias. A control group without chronic pain would more accurately represent the general healthy population and allow for a more appropriate comparison with the patient groups.”
Reviewer’s comment
- Exclusion Criteria
Please clarify the exclusion criteria used in participant selection to ensure sample purity and minimize potential sources of bias.
Author’s response
We have added two sentences in the Participants subsection to better clarify the study's inclusion and exclusion criteria.
“There were no other exclusion criteria due to the cross-sectional design of the study which did not evaluate the efficacy of any intervention.
All participants had a complete blood count, performed at the terciary hospital and measured on the same equipment.”
Reviewer’s comment
- Formatting and Presentation
Expand all abbreviations upon first use in the main text for clarity.
Author’s response
We have expanded all abbreviations upon first use in the main text.
Figures and tables: Figure 1 (panels A–D) is referenced in the text but not shown in the document excerpt. Ensure that all figures are included, high-quality, well-labeled, and correctly cited in the final version.
Author’s response
Figure 1 (panels A–D) is presented in the main document following its citation in the text.
Punctuation and spacing around units (e.g., "mg/dL") must be standardized. Inconsistent formatting detracts from readability and gives the impression of a lack of attention to detail.
Author’s response
We have standardized punctuation and unit spacing (e.g., 'mg/dL') throughout the manuscript, and we apologize for the oversights identified by the reviewer.
Correct typographical errors, such as “17,.9%” in Table 1. A thorough proofreading of the entire manuscript is strongly recommended.
Author’s response
In accordance with your suggestion, we have corrected the typographical error in Table 1 and carefully proofread the entire manuscript.
Best regards,
Dra. Miriam Almirall

Reviewer 2 Report
Comments and Suggestions for Authors
The manuscript presents the results of an observational study demonstrating the use of Neutrophil-Lymphocyte Ratio as a predictor of disease severity and activity in Axial Spondyloarthritis and Fibromyalgia.
Regarding Neutrophil-Lymphocyte Ratio in fibromyalgia, few studies no older than 10 years are found, and there is a recent increase in interest in this issue. The relationship between the neutrophil-lymphocyte ratio and the severity of fibromyalgia is also of interest. I assume that the group of patients with Axial Spondyloarthritis is provided as hypothesis testing in this study because changes in the neutrophil-lymphocyte ratio in Axial Spondyloarthritis have been widely studied. The introduction presents the hypotheses supporting the choice of the three groups in the study. The introduction also discloses the interest in the study of Neutrophil-Lymphocyte Ratio in fibromyalgia. The paper fits the journal scope in the Molecular and Translational Medicine section.
The article is standardised, the design of the experiment is fully described and includes exclusion criteria for patients, but there is no formalised Conclusion section. The methods are consistent with the hypotheses presented and the design includes three groups: controls, patients with Fibromyalgia and patients with Axial Spondyloarthritis. The ethical statement includes the registration code.
References correspond to the presented topic of the manuscript and the proportion of new sources (for the last 5 years) is relatively high, no self-citations were found.
The methods are presented in a structured manner. However, in the methods it is not clear whether the data of questionnaires, body weight, CRP level and Neutrophil-Lymphocyte Ratio for observational study were obtained at the same time.
All tables and graphs are clearly structured and easy to interpret, and some suggestions for improvement are outlined below:
- Was CRP measured on the same equipment and using the same kit manufacturer?
- In materials and methods, explain whether patients gave written consent to participate in the study.
- In Table 1, indicate the units of measurement for Duration sso, Duration sdg.
- In Tables 3 and 4, state the statistical methods for calculating P value.
- Explain in Discussion the limitations associated with the methods for determining the severity/activity of Axial Spondyloarthritis and Fibromyalgia and the associated Neutrophil-Lymphocyte Ratio. Explain in the discussions why ASDAS was chosen for arthritis in the definition of NLR cut-off values for disease severity.
- It is noteworthy that you found correlations between CRP and NLR and disease severity and activity. In my opinion, you have paid little attention to these results. Add to the discussion whether you expected this result and how it might be supported by other references. Discuss whether CRP is a meaningful criterion for fibromyalgia, which is more of a heterogeneous disease.
- Add the Conclusion section.
Author Response
Response to reviewer 2.
Neutrophil-Lymphocyte Ratio in Fibromyalgia and Axial Spondyloarthritis: A Potential Biomarker for Diagnosis and Disease Activity.
Dear reviewer,
Thank you very much for your considerations and insightful comments that will help improve our manuscript.
We proceed to answer the reviewers' comments.
Reviewer’s comment
- Was CRP measured on the same equipment and using the same kit manufacturer?
Author’s response
We have added the following sentence to the Measures subsection to address the reviewer's question and provide further clarification.
“C-Reactive Protein (CRP) levels (mg/dL). Performed at the tertiary hospital, measured on the same equipment and using the same kit manufacturer. “
- In materials and methods, explain whether patients gave written consent to participate in the study
Author’s response
Following your recommendation, we have added the following sentence to the Materials and Methods section.
“Ethical considerations
The study was conducted in accordance with the Declaration of Helsinki, and approved by the Vall d'Hebron University Hospital Ethical Committee with the registration code PR(AG)337/2024.
Patient consent was waived in accordance with current legislation (Ley Orgánica de Protección de Datos Personales y Garantía de los Derechos Digitales, LOPDGDD, 3/2018), due to the observational design of the study, the impracticality of carrying it out without waiver, the security protection provided through data pseudonymization and the absence of additional risk to participants.”
- In Table 1, indicate the units of measurement for Duration sso, Duration sdg.
Author’s response
We have clarified that the units of measurement for Duration sso and Duration sdg in Table 1 are years.
- In Tables 3 and 4, state the statistical methods for calculating P value.
Author’s response
In the Statistical Analysis subsection, the methods used for comparisons between groups are specified in the following sentence.
“Group comparisons: Differences between groups were calculated using the Kruskal-Wallis test for quantitative variables and the Chi-square test or the Fisher's exact test for categorical variables (depending on expected frequencies).”
- Explain in Discussion the limitations associated with the methods for determining the severity/activity of Axial Spondyloarthritis and Fibromyalgia and the associated Neutrophil-Lymphocyte Ratio. Explain in the discussions why ASDAS was chosen for arthritis in the definition of NLR cut-off values for disease severity.
Author’s response
The Discussion section has been expanded with the following sentences addressing the stratification of disease severity/activity in Fibromyalgia and Axial Spondyloarthritis, according to FIQR and ASDAS.
“Despite the obvious limitations of activity scores, primarily the subjectivity of assessments and the influence of external factors, a standardized stratification of Fibromyalgia patients according to disease severity is essential. This allows us to investigate whether different patient subtypes, such as those with a higher inflammatory burden or more evidence of central sensitization, are more likely to have higher values of both NLR and other inflammatory biomarkers.
We used the FIQ severity categories established in the study by Bennet et al., based on 3 placebo-controlled clinical treatment trials. “
“To stratify patients according to disease activity, in our study we used ASDAS because it has demonstrated improved measurement properties compared to BASDAI in Axial Spondyloarthritis [31].”
- It is noteworthy that you found correlations between CRP and NLR and disease severity and activity. In my opinion, you have paid little attention to these results. Add to the discussion whether you expected this result and how it might be supported by other references. Discuss whether CRP is a meaningful criterion for fibromyalgia, which is more of a heterogeneous disease.
Author’s response
We have included the following paragraph in the Discussion section to address the relationship between CRP and Fibromyalgia.
“NRL was positively correlated with CRP in both diseases, although with a smaller effect size in Fibromyalgia patients (r = .14) compared to Axial Spondyloarthritis (r = .38). These findings may be attributed to the fact that Axial Spondyloarthritis is a disease in which inflammation plays a central role, widely known and extensively studied [26]. However, as previously mentioned, further research is needed to clarify the role of inflammation in Fibromyalgia. Although Fibromyalgia is a highly heterogeneous condition that has not traditionally been associated with elevated levels of acute phase reactants, a study by Singh et al. found that increased disease severity was linked to higher levels of CRP and ESR [34].”
- Add the Conclusion
Author’s response
A Conclusions section has been added to summarize the main findings of our study and their implications, in accordance with the reviewer's recommendation.
Best regards,
Dra. Miriam Almirall

Reviewer 3 Report
Comments and Suggestions for Authors
The manuscript titled “Neutrophil-Lymphocyte Ratio in Fibromyalgia and Axial Spondyloarthritis: A Potential Biomarker for Diagnosis and Disease Activity” addresses an interesting subject. However, there are several areas where the authors can enhance the quality of the manuscript:
- Expand Keywords – The manuscript currently includes only three keywords. Adding more relevant keywords will improve discoverability and indexing.
- Discuss how significantly higher NSAID use in axSpA patients (51.8%) may impact NLR comparisons (vs. FM/HC) and consider sensitivity analyses.
- Explain the rationale for using FIQR ≥59 to define "severe" FM, given the high mean FIQR (71.8) and its implications
- Statistical Method Justification: NLR distribution normality should be confirmed. If non-normal, justify using parametric tests (Pearson correlation, t-test implied for NLR comparisons) or re-analyze using non-parametric equivalents (Spearman, Mann-Whitney) for robustness.
- Emphasize that this design precludes assessing NLR as a dynamicmarker of treatment response/disease change; frame longitudinal studies as essential.
- Conclusion Section – A dedicated conclusion section should be added to summarize findings and implications.
- Expand References – The manuscript includes only 30 references. Adding more comparative studies in the discussion section would strengthen the argument.
- Grammar & Readability – The manuscript contains grammatical errors that need correction to improve clarity and readability.
Grammar & Readability – The manuscript contains grammatical errors that need correction to improve clarity and readability.
Author Response
Response to reviewer 3.
Neutrophil-Lymphocyte Ratio in Fibromyalgia and Axial Spondyloarthritis: A Potential Biomarker for Diagnosis and Disease Activity.
Dear reviewer,
Thank you very much for your considerations and insightful comments that will help improve our manuscript.
We proceed to answer the reviewers' comments.
Reviewer’s comment
- Expand Keywords – The manuscript currently includes only three keywords.
Adding more relevant keywords will improve discoverability and indexing.
Author’s response
Following the reviewer's recommendations, we added 2 keywords: biomarker, inflammation.
Reviewer’s comment
- Discuss how significantly higher NSAID use in axSpA patients (51.8%) may impact NLR comparisons (vs. FM/HC) and consider sensitivity analyses.
Author’s response
The discussion on the effects of higher NSAID use in Axial Spondyloarthritis has been expanded with the addition of the following sentences.
“In our study, NSAID consumption was significantly higher in the Axial Spondyloarthritis group compared to Fibromyalgia patients and healthy controls. This may partially explain the more pronounced NLR values observed in Axial Spondyloarthritis, aside from the disease’s inherent inflammatory activity.
In a study by Gökmen et al., involving patients with Ankylosing Spondylitis, those receiving anti-TNF therapy had lower NLR values than those treated with NSAIDs [25]. These results were likely influenced by both the greater reduction in inflammatory activity achieved with biologic therapy and the possible effects of NSAIDs on NLR.”
Reviewer’s comment
- Explain the rationale for using FIQR ≥59 to define "severe" FM, given the high mean FIQR (71.8) and its implications.
Author’s response
To use a validated FIQ cut-off value for defining severe Fibromyalgia, we based our approach on the study by Bennet et al., which analyzed data from three placebo-controlled clinical trials. Additionally, two paragraphs have been added to the Discussion section addressing the use of the FIQR for assessing Fibromyalgia severity.
“Despite the obvious limitations of activity scores, primarily the subjectivity of assessments and the influence of external factors, a standardized stratification of Fibromyalgia patients according to disease severity is essential. This allows us to investigate whether different patient subtypes, such as those with a higher inflammatory burden or more evidence of central sensitization, are more likely to have higher values of both NLR and other inflammatory biomarkers.
We used the FIQ severity categories established in the study by Bennet et al., based on 3 placebo-controlled clinical treatment trials [26].”
Reviewer’s comment
- Statistical Method Justification: NLR distribution normality should be confirmed. If non-normal, justify using parametric tests (Pearson correlation, t-test implied for NLR comparisons) or re-analyze using non-parametric equivalents (Spearman, Mann-Whitney) for robustness.
Author’s response
The Neutrophil-Lymphocyte Ratio followed a normal distribution and, therefore, the statistical methods used are appropriate.
Reviewer’s comment
- Emphasize that this design precludes assessing NLR as a dynamic marker of treatment response/disease change; frame longitudinal studies as essential.
Author’s response
The reviewer's suggestions have been incorporated in the following sentences included in the Discussion section.
“Additionally, prospective studies should examine whether changes in NLR over time correlate with symptom fluctuations and treatment response, as suggested by recent evidence showing a decrease in NLR after duloxetine therapy [10], which may provide better insight into its clinical utility.
Further longitudinal studies are needed in both diseases, Axial Spondyloarthritis and Fibromyalgia, to evaluate the NLR’s role as a dynamic biomarker of response to different therapies.”
Reviewer’s comment
- Conclusion Section – A dedicated conclusion section should be added to summarize findings and implications.
Author’s response
A Conclusions section has been added to summarize the main findings of our study and their implications.
Reviewer’s comment
- Expand References – The manuscript includes only 30 references. Adding more comparative studies in the discussion section would strengthen the argument.
Author’s response
The discussion has been expanded to incorporate the suggestions from all three reviewers, and six new bibliographic references have been added.
Reviewer’s comment
- Grammar & Readability – The manuscript contains grammatical errors that need correction to improve clarity and readability.
Author’s response
The manuscript has been reviewed again to address and correct grammatical errors.
Best regards,
Dra. Miriam Almirall
